# Implication of Irisin in Different Types of Cancer: A Systematic Review and Meta-Analysis

**DOI:** 10.3390/ijms23179971

**Published:** 2022-09-01

**Authors:** Maria Vliora, Eleni Nintou, Eleni Karligiotou, Leonidas G. Ioannou, Elisabetta Grillo, Stefania Mitola, Andreas D. Flouris

**Affiliations:** 1FAME Laboratory, Department of Physical Education and Sport Science, University of Thessaly, 42100 Trikala, Greece; 2Department of Molecular and Translational Medicine, University of Brescia, 25121 Brescia, Italy

**Keywords:** irisin, cancers, physical exercise, thermogenesis

## Abstract

Cancer is a set of diseases characterized by several hallmark properties, such as increased angiogenesis, proliferation, invasion, and metastasis. The increased angiogenic activity constantly supplies the tumors with nutrients and a plethora of cytokines to ensure cell survival. Along these cytokines is a newly discovered protein, called irisin, which is released into the circulation after physical exercise. Irisin is the product of fibronectin type III domain-containing protein 5 (FNDC5) proteolytic cleavage. Recently it has been the topic of investigation in several types of cancer. In this study, we conducted a systematic review and meta-analysis to investigate its implication in different types of cancer. Our results suggest that irisin expression is decreased in cancer patients, thus it can be used as a valid biomarker for the diagnosis of several types of cancer. In addition, our results indicate that irisin may have an important role in tumor progression and metastasis since it is involved in multiple signaling pathways that promote cell proliferation and migration.

## 1. Introduction

Cancer is the second leading cause of death, and, according to the World Health Organization, the leading cause of death before the age of 70 around the world [1]. As it is a set of diseases, cancer can affect different tissues of the human body, characterized, among other things, by irregular growth of cells, increased angiogenesis, resistance in cell death, increased proliferative activity, invasion, and metastasis [2]. The increased angiogenic activity and extended vasculature network constantly supply tumor cells with oxygen and nutrients that are crucial for their survival [2]. In addition, tumors produce a variety of soluble factors, cytokines and growth factors that act locally to sustain tumor growth and are released into the bloodstream where they can be exploited as diagnostic/prognostic biomarkers [3].

One of these cytokines is irisin, discovered in 2012, which is released into the circulation after its proteolytic cleavage from a larger transmembrane protein, fibronectin type III domain-containing protein 5 (FNDC5) [4]. When released, it binds to αv integrin receptors and initiates signaling cascades that mediate the beneficial effects physical exercise [5]. Irisin was originally described as participating in the process called browning of white adipose tissue and the induction of thermogenesis [4]. Further studies, though, have demonstrated that irisin has a protective role against pathological conditions that are accompanied by chronic inflammation [6,7]. Inflammation supports multiple cancer hallmark processes by the recruitment of inflammatory cells and the modulation of bioactive molecules in the tumor microenvironment [2]. Thus, inflammation caused by physical exercise has important roles in the prevention and management of cancer [8]. Since irisin is a myokine released during physical exercise [9], research has focused on its potential role in cancer development and progression.

Recent studies have shown a reduced irisin expression in tissues affected by cancer in comparison with healthy tissues [10,11,12,13]. Also, irisin affects the signaling pathways in several types of cancer [14,15,16]. Even though there is an increasing amount of literature about the role of irisin in the occurrence and development of tumors or cancer prognosis, the molecular mechanisms remain as yet unelucidated. As for its biological behavior, irisin may become a new prognostic maker and/or a suitable target for the development of anti cancer drugs [17].

Our incomplete understanding about the involvement of irisin in tumor progression and prognosis limits the potential of establishing a new biomarker and the development of new therapeutic treatments. In this systematic review, we glean for the first time all the information about irisin and its involvement in cancer, including in vitro and in vivo studies that were conducted upon its discovery. The primary aim of our study is to discuss the possibility of irisin being used as a therapeutic agent for different types of cancers, as well as its capability of becoming an independent prognostic factor. Moreover, our meta-analysis strongly supports the circulating irisin as a tumoral prognostic/diagnostic biomarker. Finally, we discuss how different doses of irisin can affect cancer cell viability.

## 2. Methods

To reduce bias and the likelihood of duplication and to maximize the validity of the method used, our systematic review was registered in the international prospective register for systematic reviews (PROSPERO) database, with the registration number CRD42019119590. This systematic review also fulfills the Preferred Reporting Items for Systematic Reviews and Meta-Analyses (PRISMA) checklist.

### 2.1. Searching and Quality Assessment

#### 2.1.1. Search Strategy and Selection Criteria

Following PRISMA guidelines, PubMed central, Embase and Cochrane libraries databases were searched, from the date of their inception to 21 August 2022, for studies that researched any implication of irisin in different types of cancer. Studies that included the investigation of irisin in any biological model (cell cultures, animals, and humans) in relation to any type of cancer were included. No date or study design limits were applied. The search algorithm that was used can be found in the (Appendix A). We excluded reviews, conference proceedings, editorials, magazine articles, opinion papers, letters to editor or papers that were not in the English language, but we screened the reference list of such articles and the reviewed publications to retrieve relevant papers. Across all searches we included articles that consisted of original research published in peer reviewed journals. The searching process was conducted independently by two investigators (MV and EN) and any conflicts were resolved by consensus.

The screening of the titles, abstracts, and full texts for eligibility as well as the selection of the studies included was conducted by two investigators (MV and EN). Any conflict was resolved by a referee investigator (ADF). We included all methodological designs, using healthy or non-affected control-groups and/or interventional studies that showed the function of irisin in different signaling pathways in cancer; no sample size pre-set criterion was considered for the studies included. The list of included and excluded papers is available in the Appendix A.

#### 2.1.2. Risk of Bias Assessment

The studies included were evaluated for risk of bias by MV and EK. The selected publications consisted of both in vivo and in vitro investigations. For the assessment of risk of bias and internal validity, the tool developed from the Office for Human and Animal Studies (OHAT) was used [18]. OHAT is a tool based on eleven (11) questions (corresponding to 7 categories of bias) and four different answers (“+ +” for definitely low, “+” for probably low, “-” or “NR” for probably high or not reported, and “- -” for definitely high risk of bias). For ease of comparison the scores were recorded as +3, +2, +1 and 0 respectively [19]. The maximum total score for case control and cohort studies was 21, for experimental animal studies and randomized controlled trials was 27, and for cross-sectional studies was 24.

As mentioned above, the tool is designed to assess bias of multiple study designs but not studies conducted in vitro. Despite our thorough investigation in the literature, we did not identify a tool designed for in vitro studies. Thus, we modified the OHAT tool to fit the purpose of our in vitro studies. Within this scope, we examined bias in six out of the seven categories proposed by the OHAT tool and we slightly modified the original questions to fit the purpose of in vitro studies. The scoring system was kept as in the original tool and the maximum score in the in vitro studies was 30. The full version of the tool as well as the tool which was modified for in vitro studies can be found in Appendix A.

#### 2.1.3. Data Extraction

The data extraction was performed by MV, while EN performed data extraction on a random sample (40%) of the selected publications. Any conflicts in the data extraction were resolved through consensus and supervision by a third investigator (ADF). When necessary, additional information was requested by the corresponding authors of the selected papers via email. For all studies we extracted author name(s), year of publication, type of cancer studied, analyzed samples (cell line, animal strain, or human) and health condition of participants (healthy/diseased) (in the cases of human studies). In the case of in vivo studies, the type of cancer, the type of assay that was used to measure irisin concentration and the kit or antibody that was used were recorded. For the in vitro studies, the type of cancer model, the type (modified/non modified) of irisin that was used, the dose of irisin and period of treatment were recorded (Appendix A, Appendix A).

### 2.2. Meta-Analyses

#### 2.2.1. Differences in Irisin Protein Level and Cell Viability

We performed meta-analyses to calculate the differences between irisin levels in healthy tissues and tissues affected with cancer, and the blood irisin levels between healthy and diseased individuals. For the in vitro studies we calculated the difference in cell viability between treated and untreated cells for different doses of irisin, at different time points. In the cases where exact values were not reported in the text, we used WebPlotDigitizer (v4.5, 2021) to extract the information from the available graphs [20]. Also, we calculated the differences in cell viability between treated and untreated cells, in relation to time of treatment. For cell proliferation, migration, and invasion, there were not enough data that we could use to perform meta-analysis. The doses have been categorized into three groups, namely low (<5 nM), physiological (5–10 nΜ), and pharmacological (>10 nM) dose, according to the literature [21]. Since different methods and scales were utilized in the eligible studies, we used standardized mean differences (SMDs) instead of absolute mean differences to standardize our findings to a uniform scale. For the same reason, a random-effect model was used to account for heterogeneity due to the differences in study populations, assays used, types of cancer and cell lines. All analyses were conducted using the “metafor” package in the R language (Rstudio, version 1.3.1093, PBC, Boston, MA, USA). The level of significance was set at an alpha level of *p* < 0.05.

#### 2.2.2. Rating of Overall Study Effect

To rate the overall study effect of the studies included in our systematic review, we used the Rating of Overall Study Effect (ROSE, www.rose.reviews, accesses on 1 August 2022), which is a newly developed meta-analysis tool. In this type of meta-analysis, each article receives a score based on the study’s qualitative or quantitative data, which is then adjusted based on the number of the participants or replications of each in vitro experiment. The final ROSE score for all the assessed publications varies from −3.5 (high negative effect) to 3.5 (high positive effect). The studies were independently assessed by two independent investigators (MV and EK) based on qualitative or quantitative data. We performed two ROSE meta-analyses, one for each of the two different research questions that were addressed in this review: “Can irisin be used as a biomarker?” and “Does irisin have any role in the development and progression of cancer?”. The papers that provided information for both questions were evaluated in both meta-analyses using the relevant information each time.

## 3. Results

### 3.1. Screening of Publications

In total, 573 records were identified through the search that we performed in the three databases (401 from Embase, 126 from PubMed and 26 from Cochrane Library). Of these, 154 were duplicates and were removed. From the remaining 378 publications, 165 were excluded as reviews, commentaries, conference proceedings or in a language different than English. Finally, the abstracts of the remaining 254 papers were screened and 214 were found to be irrelevant to the aim of this systematic review. A total of 38 publications were included in this review while two more articles were retrieved by manual searching in the references of the screened articles or via alerts from the searched databases. A list of the included and excluded papers can be found in Appendix A. The searching procedure results are illustrated in a PRISMA flowchart (Figure 1).

Out of the 40 publications that were selected for this systematic review, 30 [11,12,13,16,22,23,24,25,26,27,28,29,30,31,32,33,34,35,36,37,38,39,40,41,42,43,44,45,46] described in vivo studies, of which 6 [12,13,22,30,36,47] contained immunohistochemistry results referring to the presence of irisin on tissue samples. The remaining 24 articles presented results referring to assays using blood sample extracts. Finally, 14 [14,15,16,30,47,48,49,50,51,52,53] articles presented results from in vitro experiments. In 4 [16,30] of the articles, both in vivo and in vitro experiments were conducted.

### 3.2. Risk of Bias Assessment

For the case-control studies, the papers included scored between 12 and 21 (out of a maximum score of 21) on the OHAT risk assessment. For 11 out of the 18 studies in this category, detection bias was recorded, and we could not be confident about the outcome assessment. Moreover, other sources of bias were also detected such as groups with low number of participants or exogenous reasons for cancer development. For the rest of the study designs (cohort studies, experimental animal studies, cross sectional studies and randomized controlled trials), the articles scored as “definitely low risk of bias”.

Out of the 25 assessed studies, 20 (80%) scored a probably low or definitely low risk of bias on the complete reporting item. Overall, the studies included in this review present a moderate risk of bias. The full scoring of the included papers can be found in Appendix A, Appendix A.

### 3.3. Data Extraction

#### 3.3.1. In Vivo Studies

In total, 4,357 human participant samples were investigated in the 30 publications referring to in vivo studies and included in this systematic review, of which 988 (23.14%) belonged to healthy individuals and 3349 (76.86%) to patients diagnosed with different types of cancer. Also, one publication with 60 animals (mice) was included in this systematic review, of which 12 were healthy and 48 were cancer induced (Appendix A, Appendix A).

Irisin serum level was decreased in cancer patients in 63% of the studies. In the rest of the studies, irisin was found unaltered or there was no control group to compare. One study reported an increased irisin serum protein level in renal cancer. In the in vivo study that was performed on laboratory animals (mice), irisin serum protein level was increased. (Appendix A, Appendix A). Eleven types of cancer were investigated. Interestingly, out of the 24 studies that assessed irisin protein level in serum, 12 different ELISA kits were used for producing the results (Appendix A, Appendix A).

Of the 6 studies that used immunohistochemistry to locate irisin on tissue samples, 3 used the same antibody and one did not report the source of antibody that was used. In 5 out of the 6 studies, increased irisin immunoreactivity was reported in the specimens affected by cancer (Appendix A, Appendix A).

#### 3.3.2. In Vitro Studies

The subjects in these studies were different cancer cell lines that were treated with various concentrations of irisin or other substances in the presence or absence of irisin. In total, 38 cell lines were used to investigate the role of irisin in ten types of cancer. Three studies revealed that irisin can increase chemosensitivity of cancer cell lines to other pharmacological treatments. Five studies used modified (glycosylated) irisin from different manufacturers for treating cells, three used non-modified irisin while the rest used anti-irisin antibodies for detection of the protein (Appendix A, Appendix A).

Ten out of thirteen studies reported the inhibiting role of irisin on tumor progression through decreased cell proliferation, migration, invasion, and viability, which are all hallmarks of cancer. Several signaling pathways were described, suggesting possible molecular mechanisms of irisin role in tumor progression. Interestingly, four studies suggested that irisin acts through modulation of PI3K/AKT signaling pathway which, in turn, can control Snail or NF-kB signaling cascades (Appendix A, Appendix A). In Figure 2. we present the signaling pathways that are affected from irisin as presented in the publications of this review.

### 3.4. Meta-Analysis Results

In total, we conducted five meta-analyses, two regarding the in vivo studies, calculating the difference in irisin concentration in serum or tissue between healthy and diseased participants. Three meta-analyses were conducted to calculate the effect of low, physiological, and pharmacological doses of irisin on cell viability.

#### 3.4.1. Standardized Mean Differences

For the in vivo studies that measured irisin concentration in plasma, our meta-analysis results showed that irisin decreased significantly (*p* < 0.01) in cancer patients (Figure 3).

In contrast, in the studies that assessed irisin concentration on tissues, irisin level was not altered (*p* < 0.01) after immunohistochemical localization (Figure 4).

When cancer cell lines were treated with low doses of irisin, cell viability tended to decrease but not significantly (*p* > 0.05). Though we observed a significant viability decrease between 24 and 48 h (*p* < 0.05), indicating that longer exposure to low doses of irisin can induce cell death (Figure 5).

With physiological and pharmacological doses, we observed a significant decrease in cell viability after treatment, at 24 h and 48 h, respectively (*p* < 0.01) (Figure 6 and Figure 7). Overall, we show that the higher the dose of irisin used, the more potent the decrease in cell viability is and, as expected, that the duration of treatment is also related to cell viability.

#### 3.4.2. ROSE Meta-Analysis

The ROSE platform was used to evaluate the overall study effect for the two research questions of this review: “Can irisin be used as a biomarker?” and “Does irisin have any role in the development and progression of cancer?”. In the first meta-analysis, 28 publications contained information that could address the research question, and the overall ROSE score was 2.46 (MV = 2.23 and EK = 2.68), indicating a moderately positive effect. This suggests that irisin can be used as a biomarker with moderate confidence. As for the second research question, 18 publications contained relevant information, resulting in an overall ROSE score of 2.14 (MV = 2.33 and EK = 1.95), also indicating a moderately positive effect. This suggests moderate confidence that irisin has a role in the development and progression of cancer.

Authors should discuss the results and how they can be interpreted from the perspective of previous studies and of the working hypotheses. The findings and their implications should be discussed in the broadest context possible. Future research directions may also be highlighted.

## 4. Discussion

Sedentary life, obesity and poor nutrition are believed to increase the risk for developing cancer [57,58]. The mechanisms underlying this relationship, though, have not been established yet [59]. Exercise has been linked with reduced risk of cancer development and tumor progression [60]. The effects of exercise in chronic inflammation, cytokine production and immune system can mediate the positive effects of exercise on cancer [61]. Irisin is one of the cytokines that are produced during exercise and an increasing amount of literature indicates that it may have an important role in the prognosis and treatment of various types of cancers [62,63]. Our findings in this review confirm that irisin could be used as a biomarker and/or a prognostic factor for cancer. For a prognostic biomarker, multivariate statistical analysis is required to be established as independent from other factors [64]. In a few of the studies that were assessed in this systematic review [30,31,32,33] irisin seems to be able to diagnose cancer independently of other biomarkers, making it a promising independent prognostic factor. However, since the production or targeting of specific tissues is different, unfortunately this cannot be applied to all types of cancer at present. In our review we also show that irisin is implicated in tumor progression through different signaling pathways.

The in vivo studies that assessed irisin levels in healthy and diseased patients show that irisin significantly decreases in patients with prostate [24], osteosarcoma [25], bladder [28], breast [33], colorectal [40], liver [38], and gastric cancers [35]. In addition, our meta-analyses performed by applying a random effects model confirm that overall irisin decreases in patients with cancer independently of the type of cancer or the assay that was employed to detect irisin in serum samples. These indicate that irisin could be a good diagnostic factor for cancer.

The fact that irisin is decreased in different types of cancer may indicate the interactive relationship of irisin and cancer cachexia. Cancer cachexia is a wasting syndrome that is accompanied with weight and muscle loss [65]. The loss of muscle mass can lead to a decreased amount of irisin in the circulation or in the affected tissues [66]. Irisin, on the other hand, can increase the expression of uncoupling protein 1, a protein that increases thermogenesis and energy consumption in adipose tissue [67]. Indeed, Castro et al. [27] showed that changes in myokines may induce tumor evasion and inflammation.

Furthermore, in our systematic review we also show that irisin is involved in different signaling pathways that may reveal the mechanisms through which it may be implicated in the different types of cancer. A hallmark of cancer is the endothelial to mesenchymal transition (EndMT), which is the phenotypic conversion of endothelial to mesenchymal cells and is involved in the regulation of processes such as tissue fibrosis, promotion of metastasis and chemotherapy resistance [68]. Irisin was shown to inhibit EndMT through the modulation of STAT3/Snail pathway in osteosarcoma [14] and by mediating the PI3K/Akt/Snail signaling pathway [52]. PI3K/Akt signaling pathway is also downregulated in pancreatic and liver cancer [15,16,23] by induction of cell apoptosis by irisin. Moreover, Zhang Z. et al. [39] showed in their study that serum irisin is decreased in breast cancer patients with spinal metastasis compared to non-metastatic patients. Finally, irisin was shown to inhibit NF-kB signaling by increasing chemosensitivity and inducing cell apoptosis [15] which prevents tumor progression. Also, as we show in our meta-analysis, cell viability is decreased in the presence of irisin, which makes irisin a promising agent that could be used in the treatment of cancer.

### Strengths and Limitations

To the best of our knowledge this is the first systematic review with meta-analysis conducted about the implication of irisin in different types of cancer. In our analysis we gathered and analyzed information from 34 studies, including information about 13 types of cancer and multiple subtypes of them. Also, our meta-analysis included information for 34 different cell lines that were employed in these studies. We included both in vivo and in vitro studies in our analysis to investigate both the possibility of irisin being a valid biomarker for cancer diagnosis and the effect of irisin in cancer development.

We cannot disregard the fact that in our meta-analysis we included rather diverse information, such as type of cancer, different assays for detecting irisin or measuring cell viability, different antibodies, and different types of irisin (modified and non-modified). We tried to overcome these differences by applying a random effects model in our analysis. In this review, since the goal was to identify the role of irisin in different types of cancer, we examined all the different studies using the same method of analysis. However, it would be valuable for future studies to identify patterns in irisin expression in specific cell lines or tissues, to improve our understanding of its potential for preventing or diagnosing specific types of cancer.

Nevertheless, we could not overcome limitations such as the lack of clinical history of the patients, the lack of correlation with other health and lifestyle conditions, the gender of the participants or small sample size in some of the studies, and the use of only animal models or in vitro experiments to produce the results in the eligible studies. Thus, in future studies these factors should be considered in order to create targeted, personalized treatment for patients with different characteristics.

Although in this review we collectively present the signaling pathways that are triggered by irisin and are involved in tumor progression, we still do not know what the exact stimulus is that can lead to its release into the circulation. It is well known that irisin is released after cleavage from the skeletal muscle as a response to exercise but the exact type or duration of physical activity that is required is still unknown [9]. Also, there is growing evidence that not only skeletal muscle is responsible for irisin production [69]. In our review, we show that irisin can have a protective role against cancer development. This highlights the need to shed light on the mechanisms that underlie the release of irisin in the circulation in future studies.

## 5. Conclusions

We conclude that irisin decreases in cancer patients and can be used as a valid biomarker for diagnosis of several types of cancer. Also, irisin can inhibit cell viability in a time- and dose-dependent manner, which makes it a good candidate for the development of treatments against cancer. Finally, irisin can inhibit EndMT and modulate tumor progression by its implication in multiple signaling pathways.

## Figures and Tables

**Figure 1 ijms-23-09971-f001:**
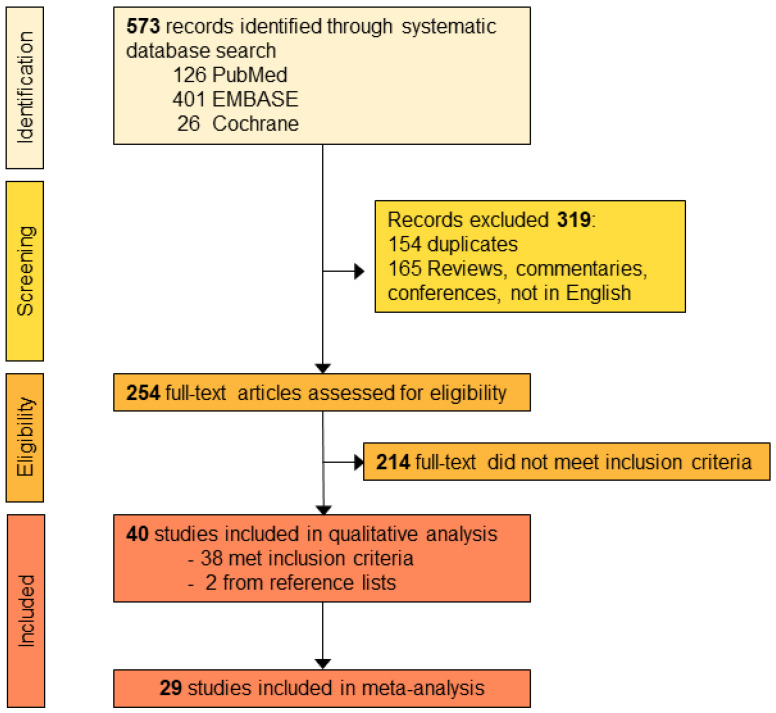
PRISMA flowchart. Selection process of papers included in the systematic review.

**Figure 2 ijms-23-09971-f002:**
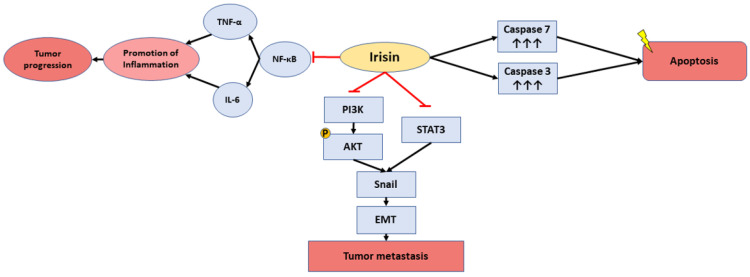
Signaling pathways that are affected by irisin in different types of cancer.

**Figure 3 ijms-23-09971-f003:**
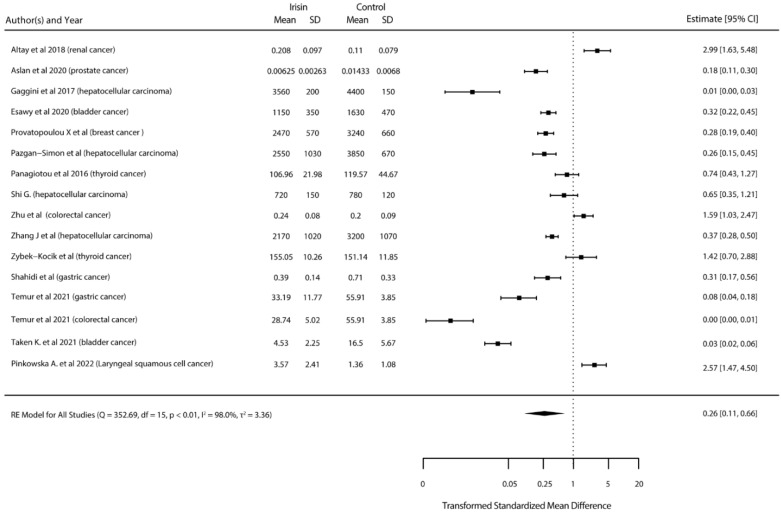
Standardized mean differences between serum irisin level in healthy and diseased participants [11,16,23,24,28,32,33,35,38,40,41,44,45,54].

**Figure 4 ijms-23-09971-f004:**
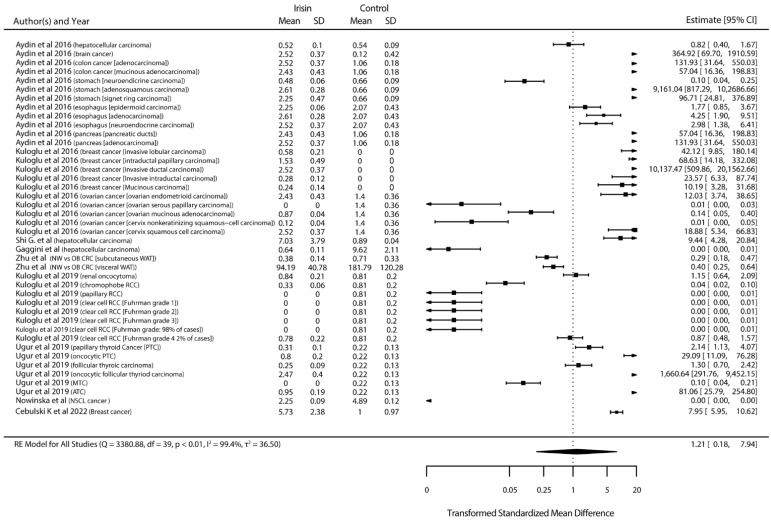
Standardized mean differences between irisin protein level on healthy and diseased tissues. (Squares indicate the mean difference of each result in reference to the scale at the bottom. Triangles are used when lower or upper bound are lower or higher than the limits of the scale used in the bottom. Diamonds indicate the overall mean difference between healthy and diseased.) [11,12,13,16,22,30,36,40,42].

**Figure 5 ijms-23-09971-f005:**
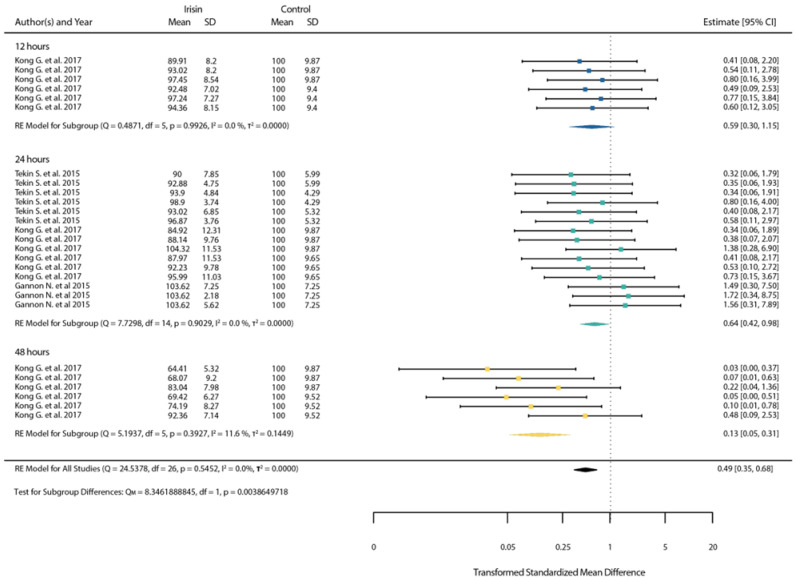
Standardized mean differences between treated and untreated cells with a low dose of irisin after 12, 24 and 48 h (shown in blue, green and yellow color respectively). (Squares indicate the mean difference of each result in reference to the scale at the bottom. Triangles are used when lower or upper bound are lower or higher than the limits of the scale used in the bottom. Diamonds indicate the overall mean difference between healthy and diseased.) [14,48,55].

**Figure 6 ijms-23-09971-f006:**
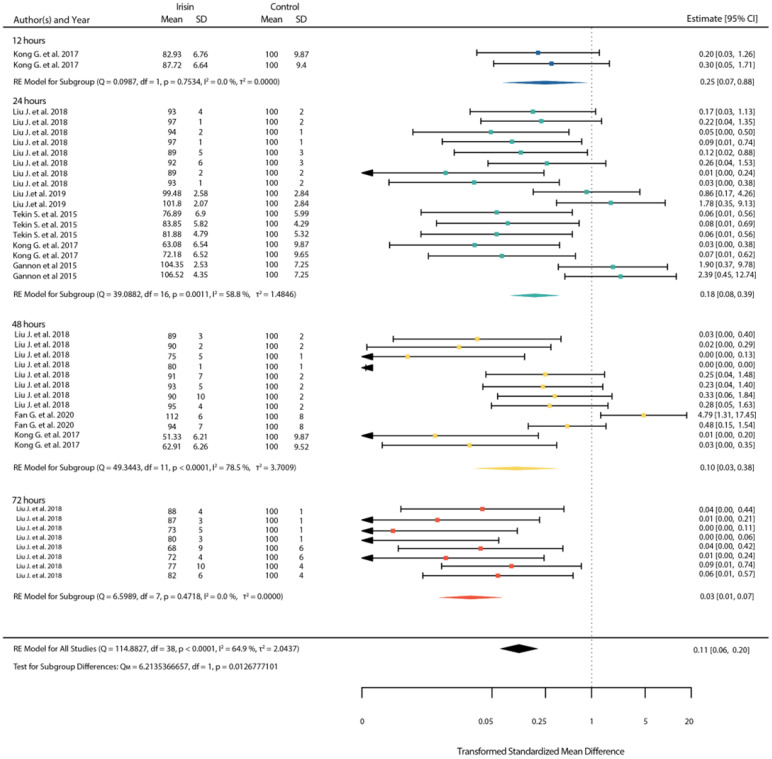
Standardized mean differences between treated and untreated cells with physiological dose of irisin after 12, 24, 48 and 72 h (shown in blue, green, yellow and red color respectively). (Squares indicate the mean difference of each result in reference to the scale at the bottom. Triangles are used when lower or upper bound are lower or higher than the limits of the scale used in the bottom. Diamonds indicate the overall mean difference between healthy and diseased.) [14,15,48,50,55,56].

**Figure 7 ijms-23-09971-f007:**
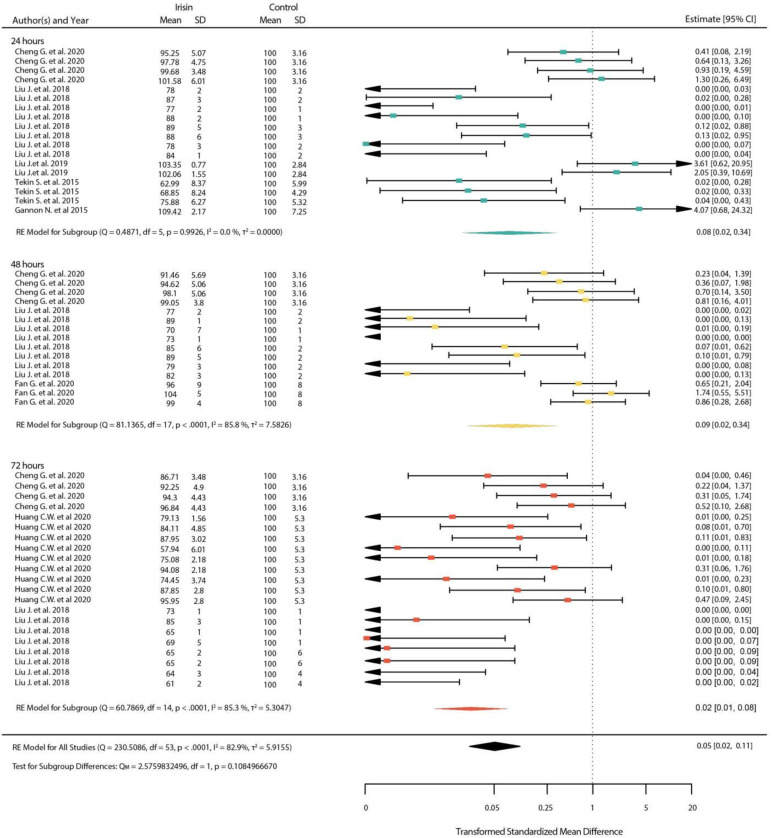
Standardized mean differences between treated and untreated cells with pharmacological dose of irisin after 24, 48 and 72 h (shown in green, yellow and red color respectively). (Squares indicate the mean difference of each result in reference to the scale at the bottom. Triangles are used when lower or upper bound are lower or higher than the limits of the scale used in the bottom. Diamonds indicate the overall mean difference between healthy and diseased.) [25,47,48,49,50,55].

## Data Availability

All data concerning this manuscript are available in the main text and the Appendix A.

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
