# Peer review of "Implication of Irisin in Different Types of Cancer: A Systematic Review and Meta-Analysis"

_ijms, 2022, doi:10.3390/ijms23179971_

Round 1
Reviewer 1 Report
The paper by Vliora M et al. summarized studies about relationship between irisin and various kinds of cancers. The antitumor effect by irisin has been reviewed in some studies (PMID: 31815594, PMID: 34071869, PMID: 34204674), but new publications are included in this paper. Moreover this is the first systematic review with meta-analysis conducted about the implication of irisin in different types of cancer both in vivo and in vitro studies. The study is well designed and the manuscript is well organized and well written and fairly easy for the reader to follow. Although there are several typos and the narrative is a kind of fragmentary, I just abide by scientific soundness. I would like to offer the following points for consideration by the authors towards the improvement of the manuscript:
1- Since there are many studies with conflicting results in the last 1 year, it would be good to update the search range and repeat the analysis.
( - https://link.springer.com/article/10.1007/s40487-022-00194-4#citeas
- https://www.ncbi.nlm.nih.gov/pmc/articles/PMC8765151/
- https://www.ncbi.nlm.nih.gov/pmc/articles/PMC8998925/
- https://link.springer.com/article/10.1007/s11255-021-03074-4
2- It is recommended that at least some meta-analyses be included in the manuscript rather than in the supplement.
3- The Discussion section could be improved. Future perspectives need to be included; what are the most 'hot topics' of current relevant research? Where do the authors believe that most future research efforts will focus on? Are there any unexplored fields that still need to be examined? What are some limitations that prevent truly personalized treatment? These are some indicative questions for which the Authors could provide their critical commentary, so that readers could gain a holistic point of view when reading the conclusion of such a broad topic review.
4- If contradictory results obtained with serum irisin determination method, cancer type, different antibodies and different irisin species cannot be included in the meta-analysis, it should be mentioned in more detail in the discussion. It may also suggest that irisin effects are cell and tissue specific.
5- Since the primary aim of the study is to discuss the possibility of irisin being used as a therapeutic agent for different types of cancers as well as its capability of becoming an independent prognostic factor it would be good if the authors would elaborate this further in the results and discussion .
Author Response
Dear Reviewers,
We wish to thank you for reviewing our work. We understand that the limitations identified during the review process were important and we appreciate the fact that you valued the interest of our paper and have given us a chance to revise it.
We made appropriate changes in the manuscript based on the comments that we received, and we believe that the revisions have markedly improved the paper’s quality.
Our point-by-point responses to the Reviewers’ comments appear below. Red font is used in the attached form to indicate revised parts of the text.
Since the present is a revised version of a previously submitted manuscript, we submitted a copy of the paper where the changes/modifications based on the comments appear as track changes.
Reviewer 1
The paper by Vliora M et al. summarized studies about relationship between irisin and various kinds of cancers. The antitumor effect by irisin has been reviewed in some studies (PMID: 31815594, PMID: 34071869, PMID: 34204674), but new publications are included in this paper. Moreover this is the first systematic review with meta-analysis conducted about the implication of irisin in different types of cancer both in vivo and in vitro studies. The study is well designed and the manuscript is well organized and well written and fairly easy for the reader to follow. Although there are several typos and the narrative is a kind of fragmentary, I just abide by scientific soundness. I would like to offer the following points for consideration by the authors towards the improvement of the manuscript:
Comment 1
- Since there are many studies with conflicting results in the last 1 year, it would be good to update the search range and repeat the analysis.
- https://link.springer.com/article/10.1007/s40487-022-00194-4#citeas
- https://www.ncbi.nlm.nih.gov/pmc/articles/PMC8765151/
- https://www.ncbi.nlm.nih.gov/pmc/articles/PMC8998925/
- https://link.springer.com/article/10.1007/s11255-021-03074-4
Response to comment 1
We would like to thank you for this constructive comment. Indeed, our previous search contained studies that have been published up until August 2021 and as it seems there have been a lot of publications during the past year that could be recognised by the searching algorithm we had used. Thus, after your suggestion we have run our search again until the date of 21st August 2022 to have an updated view of the literature.
The new papers that were added to our systematic review and meta-analysis can be found in the updated PRISMA flowchart (Figure 1) that has been added in our manuscript and the full list of excluded papers can be found in the Supplement 2_R1 file. According to the new searching results the number of papers that meet the inclusion criteria for our review now is 40 instead of 34 that was before, which include among others, the suggested publications that you have mentioned above. Accordingly, we have now changed the results in our Results section, and they can be found in our manuscript highlighted with track changes. Also, you can find the required changes in our Supplement 1 again in track changes (Tables S1-S6). Likewise, the results of our meta-analysis were changed to include the results of the studies that could fit in our meta-analysis (Figures 3-7 now in the main text).
The added studies not only did not change our previous conclusions, but they enhanced our results about irisin and its effect on different types of cancer and its potential use as a prognostic/diagnostic marker for cancer. Thus, we would like to thank you once again for your urge to update our search.
Comment 2
-It is recommended that at least some meta-analyses be included in the manuscript rather than in the supplement.
Response to comment 2
We agree with your suggestion. We have now moved the results of our metanalysis to the main text, so they are easily accessible. (Figures 3-7)
Comment 3
- The Discussion section could be improved. Future perspectives need to be included; what are the most 'hot topics' of current relevant research? Where do the authors believe that most future research efforts will focus on? Are there any unexplored fields that still need to be examined? What are some limitations that prevent truly personalized treatment? These are some indicative questions for which the Authors could provide their critical commentary, so that readers could gain a holistic point of view when reading the conclusion of such a broad topic review.
Response to Comment 3
Thank you for your comment. To address the questions that have asked in this comment we have now added the following parts in our discussion:
Line 349-359: “…to create targeted, personalized treatment for patients with different characteristics.
Although in this review we collectively present the signaling pathways that are triggered by irisin and are involved in tumor progression, we still do not know the exact stimulus that can lead to its release in the circulation. It is well known that irisin is released after cleavage from the skeletal muscle as a response to exercise but the exact type or duration of physical activity that is required is still unknown [9]. Also, there is growing evidence that not only skeletal muscle is responsible for irisin production [61]. In our review we show that irisin can have a protective role against cancer development. This highlights the need to shed light on the mechanisms that underlie the release of irisin in the circulation urgent, in future studies.”
Comment 4
- If contradictory results obtained with serum irisin determination method, cancer type, different antibodies and different irisin species cannot be included in the meta-analysis, it should be mentioned in more detail in the discussion. It may also suggest that irisin effects are cell and tissue specific.
Response to comment 4
Thank you for your comment. Even though in our review we tried to bring together the results of all the different studies, we indeed must mention that the effect of irisin in different tissues or cells can be diverse. To address this matter, we have now added the following in the discussion section:
Line 334-337: In this review, since the goal was to identify the role of irisin in different types of cancer we examined all the different studies under the same analysis. It would be valuable for future studies to identify patterns in irisin expression in specific cell lines or tissues to improve our understanding of its potential for preventing or diagnosing specific types of cancer.
Comment 5
- Since the primary aim of the study is to discuss the possibility of irisin being used as a therapeutic agent for different types of cancers as well as its capability of becoming an independent prognostic factor it would be good if the authors would elaborate this further in the results and discussion.
Response to comment 5
Thank you for your comment. We agree that the possibility of using irisin as an independent prognostic factor should be discussed more in the text. Thus, in the discussion section we have added the following:
Line 291-296: “For a prognostic biomarker to be established as independent from other factors, multivariate statistical analysis is required [62]. In few of the studies that were assessed in this systematic review [30-33] irisin seems to be able to diagnose cancer independently of other biomarkers, making it a promising independent prognostic factor. However, since the production or targeting of specific tissues is different, unfortunately this cannot be applied to all types of cancer at present.”
Reviewer 2 Report
The Authors of ijms-1872725 manuscript entitled “Implication of irisin in different types of cancer: A Systematic Review and Meta-analysis” performed very well planned and performed systematic review and meta-analysis regarding meaning of irisin in cancer development. The increasing interest in irisin is observed and documented with the increasing number of papers published a year (2012- 15 paper, 2021 – 303 papers). Searching for new biomarkers and new potential targets in cancer treatment is still ongoing trend and it is noteworthy to check the possible utility of irisin as such factor. The Authors performed systematic review of published papers according to PIRSMA. They also checked their quality with OHAT tool or with modified OHAT tool (for in vitro studies). They also used ROSE for meta-analysis performing, which is quite interesting. The whole manuscript is well constructed and informative. All steps of the experiment are well documented. In my opinion it can be interesting for the Readers to see the Figures S1-S5 in the main text of the manuscript, so I recommend to include them in the main text. I recommend publication of this paper after this minor revision.
Round 2
Reviewer 1 Report
I am satisfied that the authors have addressed all of my previous concerns about the article. It is now much improved and I feel that it is now suitable for publication.